# Frost-Resistant Rapid Hardening Concretes

**DOI:** 10.3390/ma16083191

**Published:** 2023-04-18

**Authors:** Ilyas Abdraimov, Bakhadyr Kopzhassarov, Inna Kolesnikova, Daniyar Akbulatovich Akhmetov, Ilnura Madiyarova, Yelbek Utepov

**Affiliations:** 1Department of Construction and Building Materials, Auezov University, 160012 Shymkent, Kazakhstan; 2Faculty of Construction Technologies, Infrastructure and Management, Kazakh Leading Architecture and Civil Engineering Academy, 050043 Almaty, Kazakhstan; 3Department of Construction and Building Materials, Satbayev University, 050013 Almaty, Kazakhstan; 4NIISTROMPROJECT (LLP), 050013 Almaty, Kazakhstan; 5Department of Civil Engineering, L.N. Gumilyov Eurasian National University, 010008 Astana, Kazakhstan

**Keywords:** concrete strength, microsilica, calcium chloride, hyperplasticizers, performance characteristics, rapid hardening concrete, frost resistance

## Abstract

This article presents the results of research conducted to determine the possibility of accelerating strength gain and enhancing the operational reliability of concrete. The study tested the effects of modern modifiers on concrete in order to select the composition of rapid hardening concrete (RHC) with better frost resistance characteristics. A basic composition of RHC grade C 25/30 was developed using traditional concrete calculations. Based on the analysis of previous studies by other authors, two basic modifiers (microsilica and calcium chloride (CaCl_2_)) and a chemical additive (a hyperplasticizer based on polycarboxylate esters) were selected. Then, a working hypothesis was adopted to find optimal and effective combinations of these components in the concrete composition. In the course of experiments, the most effective combination of additives for obtaining the best RHC composition was deduced by modeling the average strength values of samples in the early ages of curing. Further, RHC samples were tested for frost resistance in an aggressive environment at the ages of 3, 7, 28, 90, and 180 days to determine operational reliability and durability. The test results showed a real possibility of accelerating concrete hardening by 50% at the age of 2 days and achieving up to 25% strength gain by using both microsilica and calcium chloride (CaCl_2_). The best frost resistance indicators were observed in the RHC compositions with microsilica replacing part of the cement. The frost resistance indicators also improved with an increase in the amount of microsilica.

## 1. Introduction

Modern construction requires accelerated curing of concrete, especially in monolithic construction [1]. 

In monolithic house-building, the main economic indicator is the reduction of time needed for the construction of one floor to 3 days. This technical requirement is dictated by commercial necessity since the rapid attainment of the design strength of monolithic structures is necessary to increase the turnover of molds and formwork, make more efficient use of equipment, and enhance the productivity of construction activities. Such fast construction pace can be achieved by using concrete with accelerated strength gain, commonly known as rapid hardening concrete (RHC) [2,3]. The key advantage of RHC is its high strength in the early ages of curing, usually within the first few days. This type of concrete is suitable for situations that require the installation of a sturdy structure in a short time [4].

To accelerate the hydration of clinker minerals and intensify the concrete curing process, several technological approaches can be used. At low water–cement (W/C) ratios, there is an oversaturation of the aqueous medium by the products of hydration and hydrolysis, i.e., cement minerals, which causes accelerated strength gain [5]. The most effective strength gain occurs in well-compacted mixtures with low W/C ratios, where all the processes occur in thin layers of cement dough. Active mineral additives, such as microsilica, significantly affect the physical and technical properties of concrete, leading to improved characteristics and more efficient use of the chemical energy of clinker by creating additional centers of crystallization [6,7]. The chemical activity of these pozzolanic-based additives affects cement hydration processes (Ca(OH)_2_ + SiO_2_ + mH_2_O = CaO*SiO_2_*nH_2_O) [8,9].

To obtain high-strength RHC, the pore size can be reduced by adding ultradispersed active silica, a waste product of ferrosilicon production [10]. The use of polycarboxylate-based hyperplasticizer additives is also of interest in terms of accelerating concrete hardening. To ensure rapid hardening, an additive based on polycarboxylates should be used, as its molecules leave part of the cement surface vacant, providing free access to water by the cement grain and accelerating cement hydration [11]. In this case, an attempt can be made to simultaneously increase the amounts of polycarboxylate-based hyperplasticizer additive and cement in order to accelerate strength gain during the early ages of curing (2–7 days). Another option to accelerate concrete curing is to modify it with calcium chloride (CaCl_2_), but it should be used with caution in reinforced concretes [12].

Obtaining modified RHCs of superior quality offers various benefits, including:-High technological properties of the obtained concrete mixtures by reducing their water content and increasing their homogeneity and resistance to stratification;-Acceleration of cement hydration rate at the early stage of curing, which reduces the construction time while ensuring high quality, as described by [13];-Use of effective modifiers and hyperplasticizer additives, allowing the achievement of high rates of strength gain without reducing the operational properties.

The goal of this study was to select rapid hardening concrete compositions with over 70% compressive strength gain on day 2, equivalent to grade C 25/30 strength, and frost resistance of at least 180 cycles (where 1 cycle is 1 day of freeze–thaw). 

To achieve the goal, several solutions were defined:-Calculation of RHC compositions using two modifiers (microsilica and CaCl_2_) and laboratory testing of their strength characteristics according to [14];-Reduction of the W/C ratio according to Abrams’ law of concrete strength described in [15] and confirmation of this law using polycarboxylate ester-based highly water-reducing additives;-Study the durability of the obtained RHC compositions with over 70% compressive strength gain on day 2 by testing frost resistance using the method of alternate freezing and thawing for 180 days in an aggressive environment [16].

The results of these studies have significant implications for production in construction work. The rapid hardening concretes obtained in this study will be particularly useful in construction projects in which a quick turnaround time is required, while their notable frost resistance might be important for preventing damage to concrete structures in areas with harsh winter climates.

## 2. Materials and Methods

The research was conducted in 8 stages, each of which was aimed at solving specific tasks:-Stage 1: Selection of raw materials for the study following the regulatory documents for these materials;-Stage 2: Calculation and selection of the reference composition of RHC;-Stage 3: Addition of various amounts of modifiers (microsilica and CaCl_2_) and chemical additive based on polycarboxylates to the composition of RHC;-Stage 4: Comparison of the kinetics of compressive strength gain of samples in the early ages of curing (2 and 7 days);-Stage 5: Study of the microstructure of obtained samples of RHC under the ZEISS Axio Vert.A1 microscope (JSC ZEISS, Oberkochen, Baden-Württemberg, Germany) via reflection shooting at a magnification of ×500;-Stage 6: Selection of optimum compositions in terms of early strength and cement consumption, and testing for frost resistance in 10% salt solution (NaCl) at the ages of 3, 7, 28, 90, and 180 days.-Stage 7: Comparison of compressive strength of samples obtained after frost resistance tests at the ages of 28 and 180 days;-Stage 8: Analysis and selection of the superior compositions based on the test results of the modified RHC.

Below is a description and characteristics of the raw materials used.

Table 1 shows the characteristics of the binder (cement), taken according to [17].

Table 2 shows the chemical composition of CEM I 52.5 N. The chemical composition is given according to the standard [19].

To confirm the compliance of the selected cement with the norms and requirements of [19], several tests were carried out. The methods given in standards [17] and [20] made it possible to determine the following parameters: Grinding fineness: The cement used had a grinding fineness of 94.4%.Normal density and setting time of cement dough: The cement used had a normal consistency of 27.30%. The beginning of setting occurred after 2 h 11 min, and the end of setting occurred 4 h 10 min from the moment of mixing. These values were consistent with [17,20].

The fine aggregate used was sand from the manufacturer Giyada, LLP (Almaty region, Kazakhstan). This aggregate met the requirements of the standard [21]. Table 3 shows the characteristics of the fine aggregate used.

To obtain satisfactory characteristics of the concrete mixture and the final conglomerate of rapid hardening concrete, it is necessary to use sand with a number of dust-like inclusions, the content of which does not exceed 1.5% [21]. The content of dust and clay inclusions in the sand used in this study was 1.08%. The fineness modulus (*M_f_*) of the sand used was 2.6 (shown in Figure 1). These values were acceptable according to [21] for using this aggregate in the concrete.

The crushed stone used as a coarse aggregate in this study had 5–10 and 10–20 mm fractions and was produced by Baltabay, LLP (Almaty, Kazakhstan). This aggregate was found to meet the requirements of the standard [21]. Table 4 shows the characteristics of the coarse aggregate (crushed stone) used.

According to [22], to obtain satisfactory characteristics of the concrete mixture while sieving the coarse aggregate (crushed stone and gravel) fractions of 5–10, 10–15, 10–20, 15–20, 20–40, and 40–80 mm, as well as a combined fraction of 5–20 mm, the total passing on the reference sieves should correspond to those shown in Table 5 and Figure 2 below, where d and D represent the smallest and largest nominal grain sizes in mm, respectively.

Table 6 shows the characteristics of the modifying additive (microsilica), adopted according to [8].

MCU-95 contains spherical particles with a diameter of 0.1 microns. Its bulk density ranges from 150 to 250 kg/m^3^. According to its chemical composition, MCU-95 is represented mainly by non-crystalline silica with a content of about 97%. It has a specific surface area equal to 3980 cm^2^/g, according to [23].

Table 7 shows the chemical composition of MCU-95, taken according to [8].

Comparing the chemical composition presented by the manufacturer to the quality standard [5], it followed that the content of oxides in the composition of MCU-95 was sufficient.

Table 8 shows the characteristics of the chemical additive (polycarboxylate hyperplasticizer) adopted, according to [25].

Table 9 shows the characteristics of calcium chloride (CaCl_2_), taken according to [27].

The effectiveness of additives in regulating the strength gain kinetics of concretes and mortars was evaluated, according to [26], based on the change in the strength ($∆R_{t}$, %) of the developed compositions compared to that of the reference composition at the curing ages of 1 and 28 days (Equation (1)).
(1)$${∆R}_{t}=\left(\frac{R_{t}^{ref}-R_{t}^{dev}}{R_{t}^{ref}}\right)\cdot 100\%$$
where $R_{t}^{dev}$ and $R_{t}^{ref}$—the strengths (in MPa) of the developed concrete or mortars and the reference compositions at the curing age t, respectively.

Further studies aimed to calculate the composition of RHC with Portland cement for the first test mixes and approve the reference composition. After that, experimental confirmation was used to improve the compressive strength indicators [14] in the early curing ages (2 and 7 days) of the RHC reference composition after modifying it with accelerants (MCU-95 and CaCl_2_) proposed in previous studies [29,30,31]. In this case, the mixing time of the components after adding the additives was increased to 5 min to enable even distribution by volume. The best compositions in terms of early strength gain were tested in the laboratory for frost resistance [16] within 3 to 180 days to confirm the operational resistance and durability of the selected RHC compositions. The sequence of the tests is given below.

### 2.1. Calculation and Selection of the RHC Composition

When selecting the RHC composition with Portland cement for the first test mixes and approval of the reference composition, Equations (2) and (3) [32] were used to define the strength of concrete at an age (t, day) depending on the type of cement ($f_{cm}(t)$, MPa):(2)$$f_{cm}(t)=\beta_{cc}(t)f_{cm},$$
(3)$$\beta_{cc}\left(t\right)=exp\left\{s\left[1-{\left(\frac{28}{t}\right)}^{1/2}\right]\right\},$$
where $\beta_{cc}\left(t\right)$—a coefficient that depends on the age of the concrete, t; s—coefficient that depends on the type of cement (for CEM I 52.5 N, s=0.2).

Experimental verification and correction based on the results of the preliminary calculations of the concrete composition are mandatory [32], and the following conditions must be fulfilled:-Increase in the strength of concrete at the ages of 2 and 7 days as a result of the use of a particular method of acceleration of curing;-Increase in the strength of concrete at the ages of 2 and 7 days as a result of using several methods is not a direct sum of the increases in the strength of concrete achieved by each method separately;-Rheological properties of RHC should be taken as an average within the workability range corresponding to the slump of 10–15 cm [33].

Then, based on the RHC reference composition obtained by the calculations, several experiments were carried out:-Experiments on the acceleration of concrete curing by replacing part of the cement with MCU-95 [34];-Experiments on the acceleration of concrete curing with no silica by adding more cement to the composition than in the reference composition;-Experiments on the acceleration of concrete curing by simultaneously increasing the amounts of cement and chemical additive based on polycarboxylate, AR Premium;-Experiments designed to accelerate early curing with the addition of various amounts of CaCl_2_.

After that, test samples of concrete were made from the experimental compositions, which were tested for compressive strength at the ages of 2 and 7 days. 

### 2.2. Determination of the Compressive Strength

As part of the compressive strength test, cubes with a rib length of 100 mm were molded using a mixture of the reference composition and each subsequent modified composition. Three samples were taken per test. After the samples had reached the ages of 2 and 7 days, tests were conducted at a relative humidity of at least 95% and a temperature of 20 ± 5 °C, in accordance with [14]. The compressive strength of the concrete (*R*, MPa) was calculated with an accuracy of 0.01 MPa using Equation (4).
(4)$$R=\alpha \frac{F}{A^{\prime }}$$
where *F*—breaking load, N; *A*—the area of the working section of the sample, mm^2^; *α*—scaling factor for converting concrete strength to concrete strength in cubic samples with rib size of 100 mm.

In addition, the microstructure of RHC compositions with high early strength characteristics (2 and 7 days) was studied using a microscope. After the microstructure analysis, the compositions were tested for frost resistance.

### 2.3. Studying the Microstructure of Concrete

To study the microstructure of concrete, small pieces of concrete were cut from the area of interest and prepared for microscopy. First, any loose debris or dirt from the surface of the concrete was removed. Then, a diamond saw was used to cut small pieces of concrete. Next, progressively finer grits of sandpaper were used to polish the surface of the concrete until it was smooth and flat. Finally, a metallographic polishing machine was used to polish the surface to a mirror finish.

The polished concrete samples were then mounted onto a glass slide using a small amount of adhesive, ensuring that the area of interest was in the center of the slide.

To enhance the contrast of the concrete microstructure, the sample was stained using a 1% aqueous solution of methylene blue. The stain was applied to the sample and allowed to sit for a few minutes before being rinsed off with water.

The stained concrete sample was then placed onto the stage of the ZEISS Axio Vert.A1 microscope (JSC ZEISS, Oberkochen, Baden-Württemberg, Germany). The objective lens was selected for magnification at ×500 to visualize the microstructure of the concrete. The illumination and focus settings were adjusted to obtain clear images of the microstructure, and imaging software was used to capture and analyze the images.

The images were analyzed to identify and quantify the various components of the concrete microstructure, such as the cement paste, aggregate particles, and any voids or cracks.

After examining the microstructure of the RHC samples, the compositions with the best characteristics were tested for frost resistance.

### 2.4. Determining the Frost Resistance of Concrete

The test to determine frost resistance was carried out using the second accelerated method of alternate freezing and thawing [16]. The essence of the test is that during freezing, the liquid volume expands by more than 9% [9]. In NaCl salt solution, concrete is destroyed more quickly. Therefore, instead of a 5% solution, a 10% saturated NaCl salt solution was used as a reagent to identify possible hidden defects in the concrete structure due to rapid formation of the concrete matrix skeleton.

Tests were conducted on samples (6 per cycle, i.e., ages of 3, 7, 28, 90, and 180 days) using an LGPv laboratory freezer with the Profi electronic system (JSC Liebherr, Bulle, Switzerland). The samples were saturated in 10% NaCl solution and then thawed in a 10% NaCl solution. The accelerated tests by the second method were carried out according to the conditions and modes in Table 10 and Table 11, respectively, as specified in [16]. 

During the frost resistance tests, the main indicator was the mass loss of the samples. Samples that lost more than 2% of their original weight were considered to have failed the frost resistance test and were not further tested [16].

After obtaining the data from the frost resistance tests, the dependence of the effect of MCU-95 on the frost resistance at early ages (3, 7, 28, 90, and 180 days) and the compressive strength of RHC at the ages of 28 and 180 days was determined. Based on the tests performed, comparative calculations were made, and the composition of RHC with the best performance characteristics was determined.

## 3. Results and Discussion

Table 12 shows the test composition of RHC with pure cement, obtained by calculation according to [32] (further addressed as reference composition).

Table 12 demonstrates that the primary factor contributing to strength gain following the calculation was the low W/C ratio of 0.37, which corresponded to the proportions of the standard [35].

Table 13 displays the early curing strength characteristics of the reference composition with pure cement (No. 1 *) and compositions where part of the cement was replaced by MCU-95 (No. 2–6).

Table 13 demonstrates a general tendency towards increased early strength gain when replacing part of the cement with MCU-95. Moreover, the effect increased with the amount of MCU-95, which was consistent with the results outlined in [36]. The slight strength reduction of composition No. 5 when the MCU-95 content was 1.6% could likely be attributed to human error during testing. However, it is worth noting that despite this deviation, it achieved the desired strength target of 70% on day 2 of hardening.

Figure 3 displays an image of the microstructure of the fracture of the sample of RHC containing MCU-95, captured with a ZEISS Axio Vert.A1 microscope via reflection shooting at a magnification of ×500.

Figure 3 demonstrates that the cement stone with the addition of MCU-95 had a more homogeneous structure than the cement stone without additives [37]. The addition of active SiO_2_ created conditions for the formation of a structure with the densest packing of crystals, consisting mainly of low-base calcium silicate hydroxide (Ca_5_(OH)_2_Si_6_O_16_-4H_2_O ⇒ (CaO/SiO_2_ ˂ 1.5)), which confirmed the theory presented in [38].

Figure 4 below illustrates the effect of MCU-95 on the kinetics of strength gain of RHC during the early ages of curing, where SD stands for standard deviation.

Figure 4 shows that replacing part of the cement with MCU-95 up to 50 kg in testing RHC of grade C 25/30 allowed for an increase in strength gain of 16% at the age of 2 days and 26% at the age of 7 days while maintaining all of the strength characteristics specified in [39]. It appeared that microsilica, mainly consisting of active SiO_2_, significantly affected the fundamental physical and technical properties of concrete, resulting in improved characteristics. Specifically, microsilica contributed to a more efficient use of the chemical energy of clinker by creating additional centers of crystallization. Additionally, an increase in the amount of microsilica in the composition led to an increase in the amount of stable low-base calcium silicate hydroxide (Ca_5_(OH)_2_Si_6_O_16_*4H_2_O ⇒ (CaO/SiO_2_ ˂ 1.5)), which reduced the most unstable component of cement stone—Ca(OH)_2_ crystals. This is important for producing dense and durable concrete [29].

Table 14 presents the strength characteristics of RHC during the early ages of curing for the reference composition (No. 1 *) and compositions with increased cement content in the concrete (No. 2–7).

Table 14 indicates that increasing the amount of cement to 15% only resulted in a 1% increase in strength gain at the age of 2 days and a 2% increase at the age of 7 days. These were relatively low indicators compared to the data presented in Table 13, which reflected the strength data on the 2nd and 7th days of curing of the RHC compositions containing MCU-95.

Figure 5 illustrates an image of the microstructure of a fracture of a sample of RHC made with clean cement. The image was obtained with a ZEISS Axio Vert.A1 microscope via reflection shooting at a magnification of ×500.

Figure 5 shows that the structure of cement stone without additives was heterogeneous, had a block character, and was represented by weakly crystallized interlayers of high-base calcium hydro silicates (2Ca(OH)_2_SiO_2_-H_2_O ⇒ (CaO/SiO_2_ ˃ 1.5)), which was consistent with the opinion of the authors of [6].

Figure 6 shows the relationship between the strength gain of RHC and the cement consumption, with SD standing for standard deviation.

Figure 6, showing the dependence of strength gain on cement consumption, revealed that a simple increase in the amount of cement to 15% aimed at improving the strength gain did not give a noticeable effect, as previously indicated in [29]. 

Table 15 shows the strength characteristics of RHC in the early ages of curing (2 and 7 days) for the reference composition (No. 1 *) and compositions with a simultaneous increase in the amounts of both cement and chemical additive AR Premium (No. 2–7).

Table 15 shows that increasing the amount of cement to 15% while simultaneously increasing the amount of AR Premium to 20% demonstrated good early strength gain dynamics of RHC. Thus, at the age of 2 days, the strength increased by 51%, and at the age of 7 days, it increased by 70%. This increase in early strength with the use of AR Premium was accompanied by a decrease in the W/C ratio, which was consistent with Abrams’ law of concrete strength [15]. 

Figure 7 shows the dependence of strength gain on the amount of polycarboxylate-based hyperplasticizer AR Premium, where SD stands for standard deviation.

Figure 7 shows that increasing the amount of chemical additive based on polycarboxylates by up to 20% accelerated early strength gain. This could be anticipated since the use of PCE decreased the W/C ratio in the mixture. Therefore, the fundamental law of concrete strength, which relates strength to the W/C ratio, was not violated in this case and operated similarly in RHC. Consequently, the reduction in W/C ratio led to an increase in strength characteristics, which was consistent with the findings of previous studies [11]. 

Table 16 shows the strength characteristics during the early ages of curing of RHC for the reference composition with pure cement (No. 1 *) and compositions with varying amounts of CaCl_2_ as a percentage of the cement (No. 2–5).

From Table 16, we see that adding CaCl_2_ to the concrete mixture led to a general tendency towards an increase in early strength gain, up to a certain amount equal to 2% of the weight of cement. Furthermore, the experiments conducted suggested that an increase in the amount of CaCl_2_ to 3% of the weight of cement and more was accompanied by the same dynamics of strength gain increase at the ages of 2 and 7 days as when the amount of CaCl_2_ was equal to 2%. This complemented information on the acceleration of hardening stated in [30].

Figure 8 shows an image of the microstructure of a fracture of the RHC sample containing CaCl_2_, obtained with the ZEISS Axio Vert.A1 microscope via reflection shooting at a magnification of ×500.

The calcium oxychloride compound CaCl_2_-3Ca(OH)_2_-12H_2_O shown in Figure 8 is obtained by the chemical reaction of Ca(OH)_2_ with CaCl_2_, and it can change the microstructure of concrete as well as the properties of its pore solution. The formation of calcium oxychloride (as an expansive phase) can alter the transport properties of concrete by causing pore blockage and severe failure due to internal expansion. The chemical reaction is described by the formation of a calcium oxychloride compound using the equation 3Ca(OH)_2_ + CaCl_2_ + 12H_2_O → CaCl_2_-3Ca(OH)_2_-12H_2_O. The reaction between calcium chloride and calcium hydroxide is rapid and can weaken the structure if formed in the concrete matrix due to internal hydraulic pressures that can occur. The calcium oxychloride compound is relatively unstable at room temperature and low levels of relative humidity. Calcium oxychloride can exist in different molar ratios, including CaCl_2_-3Ca(OH)_2_-12H_2_O, CaCl_2_-Ca(OH)_2_-xH2O, and CaCl_2_-Ca(OH)_2_. These phases can coexist and are interchangeable in the compound Ca(OH)_2_-CaCl_2_-H_2_O. Hence, the compound CaCl2-3Ca(OH)2-12H2O, as seen in Figure 8, is destructive and can change to CaCl_2_-Ca(OH)_2_ or CaCl_2_-Ca(OH)_2_-xH_2_O with changes in temperature or ambient humidity [12]. 

Figure 9 shows the dependence of the strength gain kinetics of RHC in the early ages of curing on the amount of CaCl_2_, where SD stands for standard deviation.

Figure 9 shows that the addition of CaCl_2_ in the amount of 2% of the weight of cement affected the curing acceleration by 9.4 MPa at the early age of 2 days and by 7.5 MPa at the age of 7 days, which was consistent with [30,31].

After comparing the data on strength during the early hardening period at 2 and 7 days, examining the microstructure of the obtained RHC samples, and studying previous works in this research area [10,11], it was decided to conduct further testing on the frost resistance of RHC samples. In this testing, some of the cement was replaced by MCU-95 microsilica. Three types of tested compositions, except for the specimens with pure cement, showed a compressive strength of 70% or more of concrete grade C 25/30 on the second day of hardening. However, only the samples with some cement replaced by microsilica had a more uniform structure compared to the others, as revealed by the microstructure study.

Table 17 presents data on the strength and weight of RHC for the reference composition with pure cement (No. 1 *) and compositions with varying amounts of MCU-95 (No. 2–6) from Table 13 after being tested for frost resistance in an aggressive environment (i.e., in 10% NaCl solution) for 3, 7, 28, 90, and 180 days.

Table 17 shows a general tendency of improved frost resistance in RHC modified with MCU-95 (No. 2–6) in comparison to the reference composition made with conventional cement (No. 1 *). Furthermore, the better the performance, the higher the content of MCU-95. This indicated that the obtained results were in compliance with the theoretical assumptions regarding the creation of additional crystallization centers and reduction of pore space in the concrete body by using reactive pozzolanic additives (active SiO_2_) [10].

Figure 10 shows the process of weight reduction of the RHC samples with different contents of MCU-95 and the RHC reference composition without MCU-95 during the test for frost resistance, where SD stands for standard deviation.

Figure 10 shows a general trend of improvement in the frost resistance index of RHC compositions modified with MCU-95 (No. 2–6) compared to the reference composition with cement (No. 1), which failed the test for frost resistance after 180 days by losing over 2% in mass, which agreed with the opinion of the authors of [40].

Figure 11 shows the process of strength reduction in RHC compositions with different content of MCU-95 (No. 2–6) and the reference composition (No. 1) without MCU-95 at the age of 28 and 180 days during the test for frost resistance in 10% NaCl.

Figure 11 shows a general trend towards improved frost resistance index in the RHC compositions modified with MCU-95 (with higher MCU-95 content leading to higher indicators) compared to the reference composition of conventional cement RHC. The reference composition failed the frost resistance test after 180 days, with a 10% loss in strength, which is consistent with the findings of [40].

Based on the data collected on the strength of RHC samples during the early hardening period at 2 and 7 days, it can be noted that almost all compositions, except those using pure cement, achieved the target of reaching a compressive strength of 70% or more of concrete grade C 25/30 at 2 days. Interestingly, increasing the amount of cement beyond the reference composition did not lead to higher early strength, which was an unexpected result. However, this finding was partially explained by previous studies [6].

The results of the compressive strength tests conducted at 2 and 7 days of hardening for the remaining three types of samples were as expected, with all of them exhibiting a compressive strength of at least 70%. The presence of active SiO_2_ in compositions with the replacement of some of the cement with microsilica facilitated the formation of an early structure with the densest packing of crystals, as evidenced by [38], which consisted primarily of low-base hydroxide calcium silicate (Ca_5_(OH)_2_Si_6_O_16_*4H_2_O ⇒ (CaO/SiO_2_ ˂ 1.5)). The use of polycarboxylate plasticizer in compositions with an increase in the amount of cement up to 15% and hyperplasticizer AR Premium up to 20% resulted in good early strength gain dynamics of RHC up to 110% at the age of 2 days, accompanied by a decrease in the B/C ratio. The addition of CaCl_2_ in the concrete mixture led to an increase in early strength gain due to early bonds between portlandite Ca(OH)_2_ and calcium chloride CaCl_2_, resulting in the formation of gel-like calcium oxychloride CaCl_2_*3Ca(OH)_2_12H_2_O, up to a certain amount equal to 2% of the weight of cement [30,31]. Further increase in the amount of CaCl_2_ had no additional effect on the strength gain at the ages of 2 and 7 days. The frost resistance tests on the samples of RHC showed that the best indicators of frost resistance were observed in the samples where the cement part was replaced by microsilica in the quantity of up to 50 kg per 1 m^3^. The performance was better with a higher content of microsilica, indicating the higher operational reliability of products made of concrete with active microsilica. The results confirmed the theory of the creation of additional centers linking hydrolyzed lime into an additional amount of strong, cementitious calcium hydrosilicates and showed the compliance of the obtained results with the theoretical assumptions about the creation of additional centers of crystallization and the reduction of pore space in the concrete body when using reactive pozzolanic additives.

## 4. Conclusions

The results of the tests confirmed the purpose of the research, which was to select RHC compositions that achieved a strength gain of 70% or more in the first 2 days of hardening. Out of the four series of samples tested, three samples with different modifier and binder contents were able to achieve the required early strength indicators.

Based on the calculations and laboratory tests, it can be concluded that the use of modified RHC is justified both from technical and scientific points of view. Compared to ordinary cement concrete compositions, RHC with modifiers can accelerate the construction process. RHC with MCU-95 microsilica accelerated the strength gain rate and exhibited the best frost and strength indicators at 90 and 180 days after placement, indicating the absence of destructive processes in concrete due to the use of microsilica and good performance characteristics. 

The test results demonstrated the consistency of the obtained data in RHC with the addition of microsilica with the theoretical assumptions regarding the emergence of additional centers of crystallization and the reduction of pore space in the concrete body during the use of reactive pozzolanic additives (active SiO_2_). The process of binding Ca(OH)_2_ by the active mineral additive SiO_2_ into the low-solubility compound—calcium hydrosilicate—occurred according to the equation: Ca(OH)_2_ + SiO_2_ + mH_2_O = CaO*SiO_2_*nH_2_O. Experiments with accelerated curing using a specific amount of CaCl_2_ as well as compositions with simultaneously increased amounts of cement and hyperplasticizer based on the polycarboxylate AR Premium demonstrated an improvement in the early strength gain of RHC. However, microscopic examination of the concrete samples revealed a heterogeneous and weakly crystallized structure, indicating the poor quality and durability of the resulting RHC, especially since chlorides should be used with caution in reinforced structures.

The compositions of RHC (No. 4–6) with MCU-95 from Table 13 that showed high strength gain at 2 and 7 days can be recommended for application in production. Furthermore, these compositions maintained their mass and compressive strength after frost resistance tests at 90 and 180 days, which is an important indicator of durability. Summarizing the foregoing, it can be said that the research team fulfilled the objectives of this study and obtained satisfactory results.

## Figures and Tables

**Figure 1 materials-16-03191-f001:**
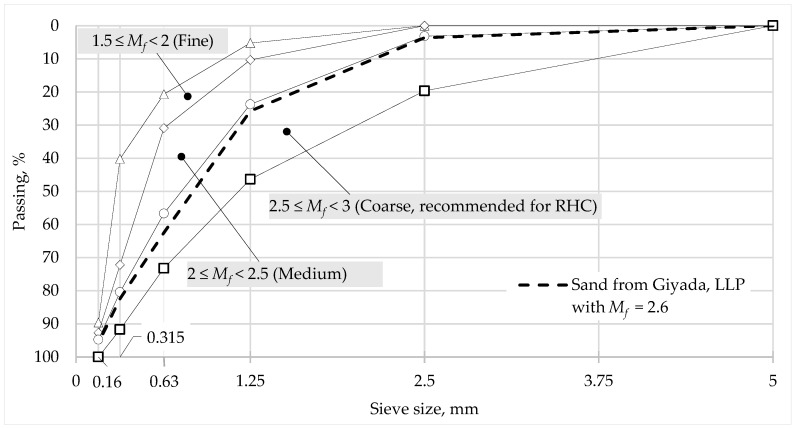
Sieve analysis grading curve for fine aggregate [22].

**Figure 2 materials-16-03191-f002:**
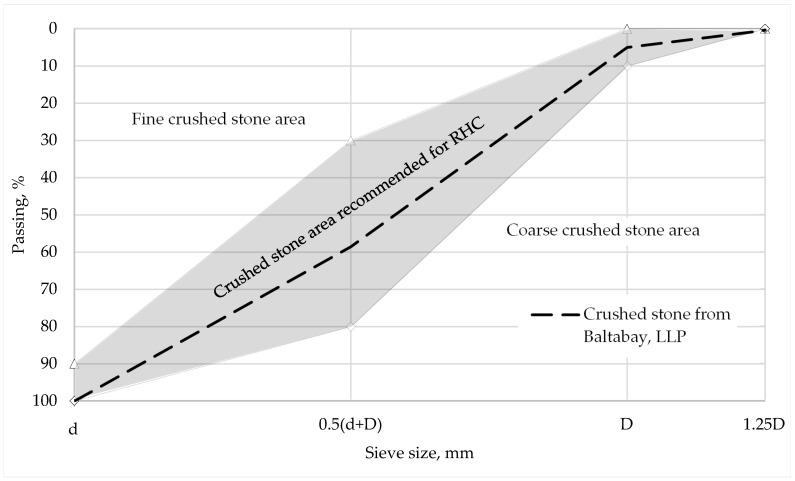
Sieve analysis grading curve for coarse aggregate [22].

**Figure 3 materials-16-03191-f003:**
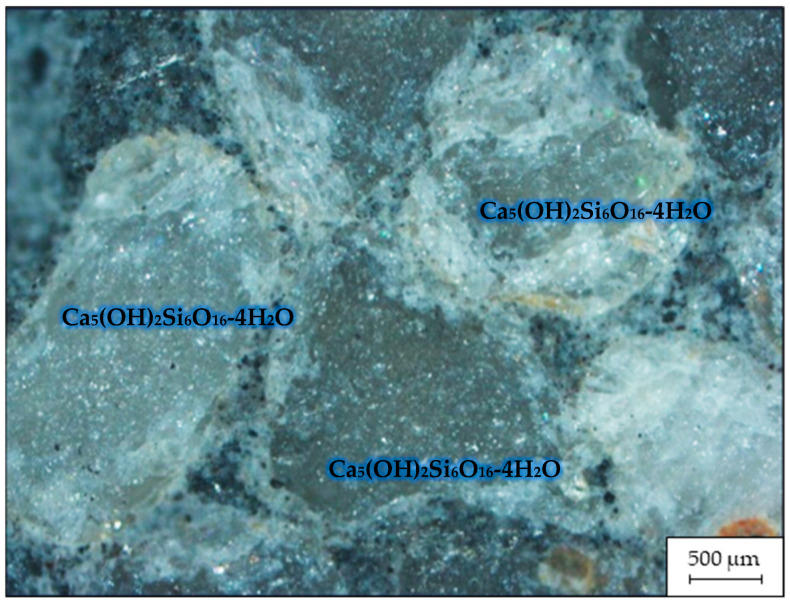
Microstructure of a fracture of the RHC sample containing MCU-95.

**Figure 4 materials-16-03191-f004:**
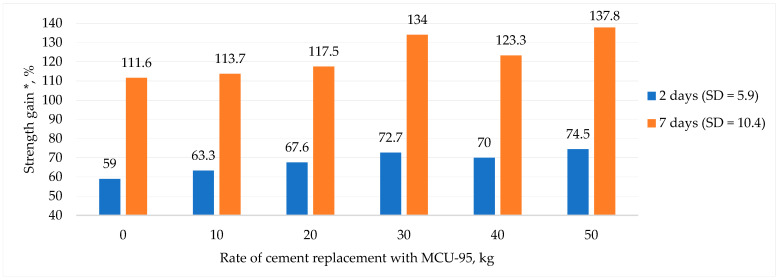
The effect of MCU-95 microsilica on strength gain. * For 100%, the average compressive strength of 30.0 MPa of the RHC reference composition of grade C 25/30 was used [35].

**Figure 5 materials-16-03191-f005:**
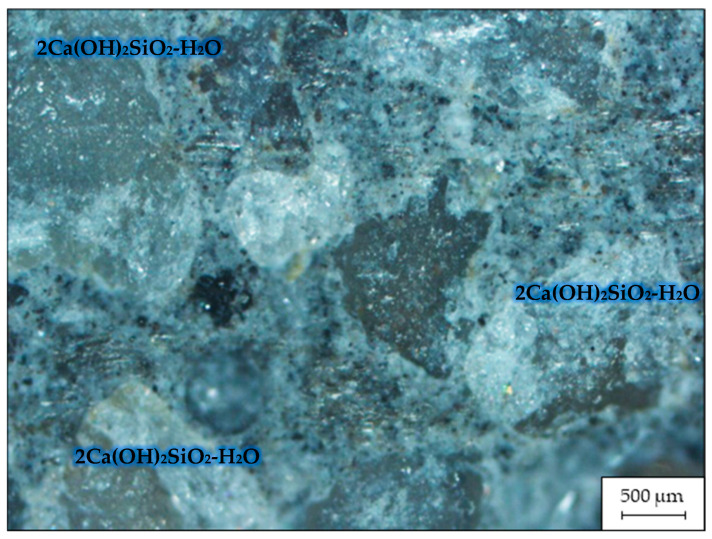
Microstructure of a fracture of the RHC with pure cement without any additives.

**Figure 6 materials-16-03191-f006:**
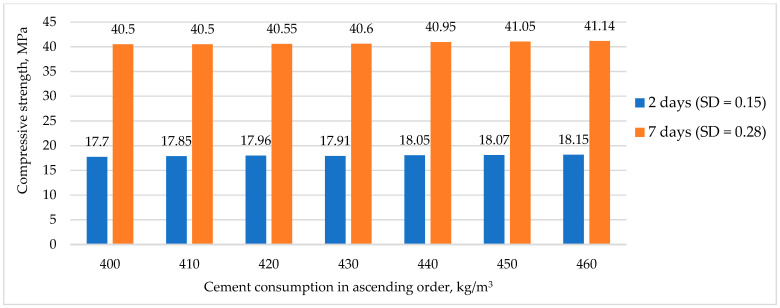
The effect of cement consumption on RHC early strength gain.

**Figure 7 materials-16-03191-f007:**
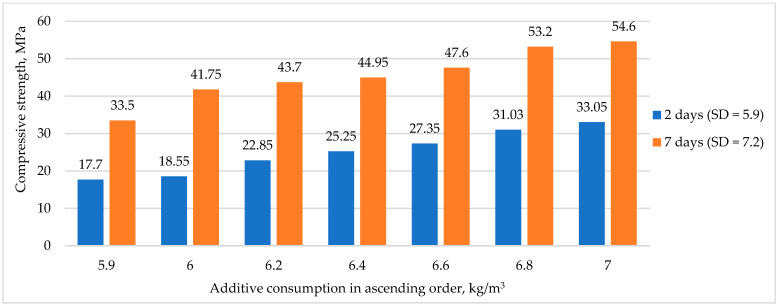
The effect of PCE hyperplasticizer amount on RHC strength gain.

**Figure 8 materials-16-03191-f008:**
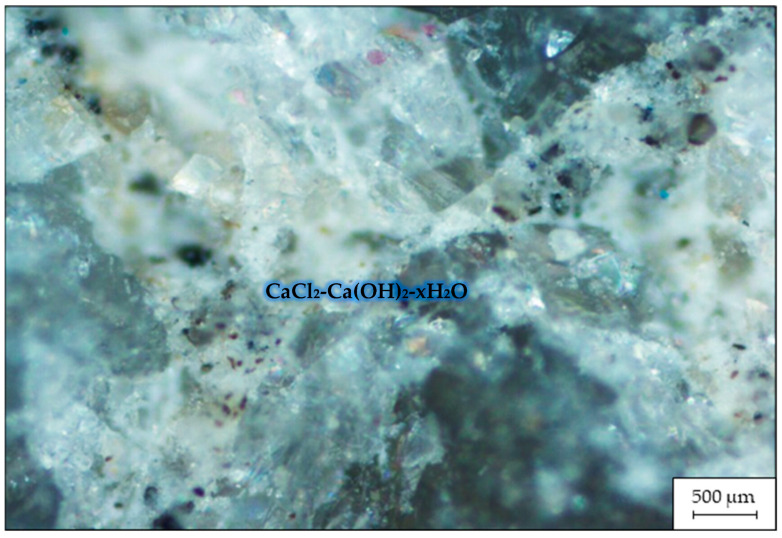
Microstructure of a fracture of the RHC containing CaCl_2_.

**Figure 9 materials-16-03191-f009:**
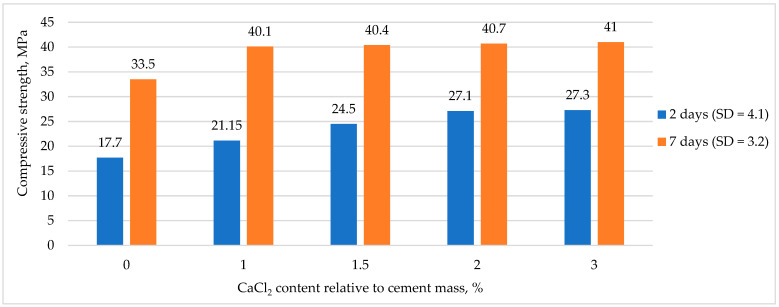
The effect of CaCl_2_ amount on RHC strength gain.

**Figure 10 materials-16-03191-f010:**
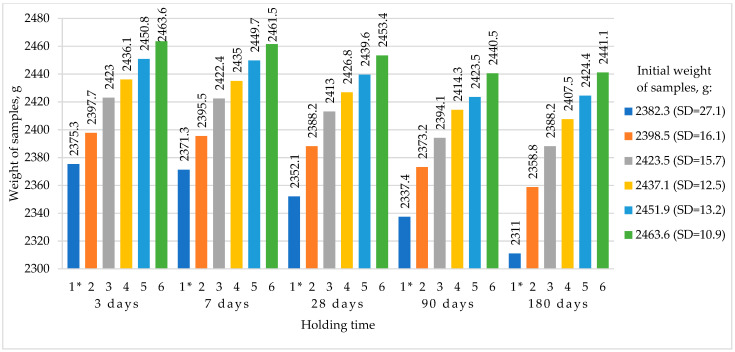
Weight reduction of RHC samples after frost resistance tests. * Reference composition.

**Figure 11 materials-16-03191-f011:**
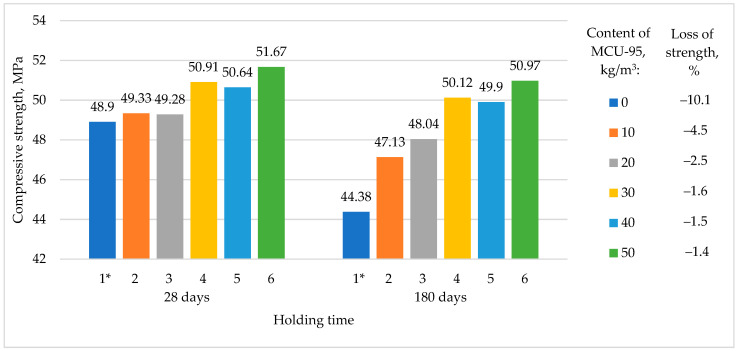
Decrease in the strength of RHC compositions after frost resistance tests. * Reference composition.

**Table 1 materials-16-03191-t001:** Characteristics of cement [17,18].

Brand Name	Manufacturer	Compressive Strength at the Age of 28 Days at Least, MPa	Start of Setting, Not Earlier than, min	Consumption per 1 m^3^ of Concrete, kg
CEM I 52.5 N *	Standard Cement, LLP (Shymkent, Kazakhstan)	60.7	105	360–460

* Ref. [17] gives the following decoding of CEM I 52.5 N: I—denotes the first type according to the content of gypsum (counted for SO_3_), which should be at least 1.5% and not higher than 3.5–4.0% (for high-strength cement without additives); 52.5—denotes the strength class corresponding to the minimum standard value of compressive strength, MPa; N—denotes the normal hydration reaction.

**Table 2 materials-16-03191-t002:** Mass fraction content of CEM I 52.5 N, % [19].

SiO_2_	Al_2_O_3_	Fe_2_O_3_	CaO	MgO	Na_2_O	K_2_O	SO_3_	Other Impurities
21.30	4.75	4.86	63.68	3.07	0.77	0.78	0.48	0.36

**Table 3 materials-16-03191-t003:** Characteristics of sand [21].

Group of Sand	Manufacturer	Fineness Modulus	Total Passing on 0.63 mm Sieve, %	Content of Dust and Clay Inclusions, %	Consumption per 1 m^3^ of Concrete, kg
Coarse	Giyada, LLP (Almaty, Kazakhstan)	2.6	62.5	1.08	800–1000

**Table 4 materials-16-03191-t004:** Characteristics of crushed stone [21].

Grain Size, mm	Manufacturer	Total Passing on Sieves 0.5 (d+D), %	Total Passing on Sieve 1.25D, %	Consumption per 1 m^3^ of Concrete, kg
5–10	Baltabay, LLP (Almaty, Kazakhstan)	57.43	0.39	200–400
10–20		59.61	0.43	500–700

**Table 5 materials-16-03191-t005:** Recommended coarse aggregate sieving parameters [22] *.

Diameter of Holes in the Reference Sieves, mm	Total Passing on Sieves, %
d	From 90 to 100
0.5 (d+D)	From 30 to 60
D	Up to 10
1.25D	Up to 0.5

* For the crushed stone and gravel fraction of 5–10 mm and combined fraction of 5–20 mm for which additionally lower sieves (2.5 or 1.25 mm) are used, the total passing must be from 95% to 100%. Upon agreement between the manufacturer and the consumer, producing crushed stone and gravel with passing on sieve 0.5 (d+D) from 30% to 80% by mass is allowed [22].

**Table 6 materials-16-03191-t006:** Characteristics of microsilica [23].

Brand Name	Manufacturer	Mass Fraction of Active SiO_2_, % by Mass, Not Less than 95%	Consumption per 1 m^3^ of Concrete, kg
MCU-95	Tau-Ken Temir, LLP (Karaganda, Kazakhstan)	96.85	up to 50

**Table 7 materials-16-03191-t007:** Chemical composition of MCU-95, % [24].

SiO_2_	C	Water	Fe_2_O_3_	Al_2_O_3_	CaO	pH	*ρ*, g/cm^3^
96.85	1.31	1.07	0.07	0.24	0.46	7.89	0.44

**Table 8 materials-16-03191-t008:** Characteristics of polycarboxylate hyperplasticizer [26].

Brand Name	Manufacturer	Criterion for Additive Effectiveness	Consumption per 1 m^3^ of Concrete, kg
AR Premium	ARPG, LLP (Astana, Kazakhstan)	from P1 to P5	from 5 to 8

**Table 9 materials-16-03191-t009:** Characteristics of CaCl_2_ [27,28].

Brand Name	Manufacturer	Freezing Temperature, °C	Particle Size, mm	Consumption per 1 m^3^ of Concrete, kg
1 type	EuroChem, LLC (Moscow, Russia)	−48	7–8	1–3

**Table 10 materials-16-03191-t010:** Test conditions for determining the frost resistance [16].

Cubic Sample Rib Size, mm	Freezing		Thawing	
	Time, Hours (Not Less than)	Temperature, °C	Time, Hours (Not Less than)	Temperature, °C
100	2.5	−18 ± 2	2 ± 0.5	20 ± 2

**Table 11 materials-16-03191-t011:** Test mode [16].

Method and Grade of Concrete by Frost Resistance	Test Conditions			Type of Concrete
	Saturation Environment	Freezing Environment and Temperature	Thawing Environment and Temperature	
Basic methods				
First, F1	Water	Air, −18 ± 2 °C	Water, 20 ± 2 °C	All types of concrete, except road and airfield pavement concretes and concretes of structures operating under the influence of mineralized water.
Second, F1	5% aqueous NaCl solution	Air, −18 ± 2 °C	5% aqueous NaCl solution, 20 ± 2 °C	Concretes of road and airfield pavements and concretes of structures operated under the action of mineralized water.

**Table 12 materials-16-03191-t012:** Reference composition of RHC grade C 25/30.

Cement		Sand		Crushed Stone (5–10 mm)		Crushed Stone (10–20 mm)		AR Premium		Water		Density
kg	%	kg	%	kg	%	kg	%	kg	%	kg	%	kg /m^3^
400	16.1	971	39.2	287	11.6	667	26.9	5.9	0.24	148	6	2382.3

**Table 13 materials-16-03191-t013:** Compositions of RHC with various contents of MCU-95 per 1 m^3^ and data on the strength at the ages of 2 and 7 days of curing.

No.	W/C	Cement		Sand		Crushed Stone (5–10 mm)		Crushed Stone (10–20 mm)		AR Premium		MCU-95		Water		Compressive Strength at the Age of 2 Days ^1^		Compressive Strength at the Age of 7 Days ^1^	
		kg	%	kg	%	kg	%	kg	%	kg	%	kg	%	kg	%	MPa	%	MPa	%
1 *	0.37	400	16.1	971	39.2	287	11.6	667	26.9	5.9	0.24	-	-	148	6.0	17.7	59.0	33.5	111.6
2	0.37	400	16.0	971	38.5	287	11.5	667	26.8	5.9	0.24	10	0.4	152	6.1	19.0	63.3	34.1	113.7
3	0.36	390	15.7	971	39.0	287	11.5	667	26.8	5.9	0.24	20	0.8	147	5.9	20.3	67.6	35.25	117.5
4	0.36	380	15.3	971	39.0	287	11.5	667	26.8	5.9	0.24	30	1.2	147	5.9	21.8	72.7	40.2	134.0
5	0.35	370	14.9	971	39.1	287	11.6	667	26.8	5.9	0.24	40	1.6	144	5.8	20.9	70.0	37.0	123.3
6	0.35	360	14.5	971	39.1	287	11.6	667	26.8	5.9	0.24	50	2.0	144	5.8	22.35	74.5	41.35	137.8

* Reference composition. ^1^ For 100%, the average compressive strength of 30.0 MPa of the RHC reference composition of grade C 25/30 was used [35].

**Table 14 materials-16-03191-t014:** Composition of RHC per 1 m³ with an increased amount of cement.

No.	W/C	Cement		Sand		Crushed Stone (5–10 mm)		Crushed Stone (10–20 mm)		AR Premium		Water		Compressive Strength at the Age of 2 Days ^1^		Compressive Strength at the Age of 7 Days ^1^	
		kg	%	kg	%	kg	%	kg	%	kg	%	kg	%	MPa	%	MPa	%
1 *	0.37	400	16.1	971	39.2	287	11.6	667	26.9	5.9	0.24	148	6	17.7	59.0	40.5	135.0
2	0.36	410	16.5	961	38.6	287	11.5	667	26.8	5.9	0.24	147.6	5.9	17.85	59.5	40.5	135.0
3	0.35	420	16.9	951	38.2	287	11.5	667	26.8	5.9	0.24	147	5.9	17.96	59.9	40.55	135.2
4	0.35	430	17.3	941	37.8	287	11.5	667	26.8	5.9	0.24	150.5	6.1	17.91	59.7	40.6	135.3
5	0.35	440	17.7	931	37.5	287	11.6	667	26.8	5.9	0.24	154	6.2	18.05	60.2	40.95	135.5
6	0.35	450	18.1	921	37	287	11.5	667	26.8	5.9	0.24	157.5	6.3	18.07	60.2	41.05	136.8
7	0.34	460	18.5	911	36.6	287	11.5	667	26.8	5.9	0.24	156.5	6.3	18.15	60.2	41.14	137.1

* Reference composition. ^1^ For 100%, the average compressive strength of 30.0 MPa of the RHC reference composition of grade C 25/30 was used [35].

**Table 15 materials-16-03191-t015:** RHC compositions per 1 m^3^ with the increased amount of cement and AR Premium.

No.	W/C	Cement		Sand		Crushed Stone (5–10 mm)		Crushed Stone (10–20 mm)		AR Premium		Water		Compressive Strength at the Age of 2 Days ^1^		Compressive Strength at the Age of 7 Days ^1^	
		kg	%	kg	%	kg	%	kg	%	kg	%	kg	%	MPa	%	MPa	%
1 *	0.37	400	16.1	971	39.2	287	11.6	667	26.9	5.9	0.24	148	6	17.7	59.0	33.5	111.9
2	0.35	400	16.2	971	39.3	287	11.6	667	27	6.0	0.24	140	5.7	18.55	61.8	41.75	139.2
3	0.34	410	16.6	961	38.9	287	11.6	667	27	6.2	0.25	139.5	5.6	22.85	76.2	43.7	145.7
4	0.33	420	17	951	38.5	287	11.6	667	27	6.4	0.26	138.5	5.6	25.25	84.2	44.95	149.8
5	0.31	430	17.5	941	38.2	287	11.6	667	27	6.6	0.27	133.5	5.4	27.35	92.2	47.6	158.7
6	0.29	440	17.9	931	37.9	287	11.7	667	27.1	6.8	0.28	127.5	5.2	31.03	103.4	53.2	177.3
7	0.28	450	18.3	921	37.5	287	11.7	667	27.1	7.0	0.29	126	5.1	33.05	110.2	54.6	182.0

* Reference composition. ^1^ For 100%, the average compressive strength of 30.0 MPa of the RHC reference composition of grade C 25/30 was used [35].

**Table 16 materials-16-03191-t016:** Compositions of RHC containing CaCl_2_.

No.	W/C	Cement		Sand		Crushed Stone (5–10 mm)		Crushed Stone (10–20 mm)		AR Premium		CaCl_2_		Water		Compressive Strength at the Age of 2 Days ^1^		Compressive Strength at the Age of 7 Days ^1^	
		kg	%	kg	%	kg	%	kg	%	kg	%	kg	%	kg	%	MPa	%	MPa	%
1 *	0.37	400	16.1	971	39.2	287	11.6	667	26.9	5.9	0.24	-	-	148	6	17.7	59.0	33.5	111.7
2	036	400	16.1	971	39.2	287	11.6	667	26.9	5.9	0.24	4.0	0.2	144	5.8	21.15	70.5	40.1	133.7
3	0.35	400	16.2	971	39.2	287	11.6	667	26.9	5.9	0.24	6.0	0.2	140	5.6	24.5	81.7	40.4	134.6
4	0.34	400	16.2	971	39.2	287	11.6	667	26.9	5.9	0.24	8.0	0.3	136	5.5	27.1	90.3	40.7	135.7
5	0.34	400	16.1	971	39.2	287	11.6	667	26.9	5.9	0.24	12.0	0.5	136	5.5	27.3	91.0	41.0	136.7

* Reference composition. ^1^ For 100%, the average compressive strength of 30.0 MPa of the RHC reference composition of grade C 25/30 was used [35].

**Table 17 materials-16-03191-t017:** Results of tests of RHC compositions containing MCU-95 for frost resistance in 10% NaCl solution.

No.	Initial Weight of Samples, g	Weight of Samples after 3 Days	Weight of Samples after 7 Days	Weight of Samples after 28 Days	Weight of Samples after 90 Days	Weight of Samples after 180 Days	Loss of Weight in %	Compressive Strength, MPa		Loss of Strength in %
								28 Days	180 Days	
1 *	2382.3	2375.3	2371.3	2352.1	2337.4	2311.0	3.0	48.9	44.38	10.1
2	2398.5	2397.7	2395.5	2388.2	2373.2	2358.8	1.7	49.33	47.13	4.5
3	2423.5	2423.0	2422.4	2413.0	2394.1	2388.2	1.5	49.28	48.04	2.5
4	2437.1	2436.1	2435.0	2426.8	2414.3	2407.5	1.2	50.91	50.12	1.6
5	2451.9	2450.8	2449.7	2439.6	2423.5	2424.4	1.1	50.64	49.9	1.5
6	2463.6	2463.6	2461.5	2453.4	2440.5	2441.1	0.9	51.67	50.97	1.4

* Reference composition.

## Data Availability

The data presented in this study cannot be shared at this time. They may be available from the corresponding author upon reasonable request.

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
