# Peer review of "Frost-Resistant Rapid Hardening Concretes"

_materials, 2023, doi:10.3390/ma16083191_

Round 1
Reviewer 1 Report (Previous Reviewer 2)
In this paper, the effects of three modifiers on the early age strength and frost resistance of rapid hardening concrete are studied. However, the analysis of the test results needs to be further deepened. The detail comments are as follows:
(1) The introduction presented now seems like the test report rather than scientific paper. Please clarify the purpose and significance of your research differing Refs.[13-16] in the last paragraph of the introduction.
(2) Figs. 3,5, and 8 are not clear enough, and please add the text label to the figures to help understand your statement. Line 286 describes the comparison of the microstructure of RHC containing MCU-95 and reference cement stone. However, only the SEM image of RHC containing MCU-95 is shown in Fig. 3, so it is necessary to provide the microscopic SEM image of reference cement stone. The SEM images of reference cement stone can be derived from experiments or references to previous research results.
(3) With the increase of MCU-95 content, the compressive strength of RHC at 2d age gradually increased, the author should try to analyze the reasons.
(4) It is suggested to analyze the change rule of RHC compressive strength at 7d age, and explain the reason of strength reduction when MCU-95 content is 1.6 %, which is the test operation or other reasons.
(5) Line 350, try to analyze the reasons for the increase of RHC strength after adding hyperplasticizer.
(6) It is not easy to find Ca(OH)2 and CaCl2*3Ca(OH)2*12H2O in Fig.8.
(7) In line 381, curing acceleration by 29 % at the early age of 2 days and by 22 % at the age 382 of 7 days, it is necessary to explain how 29 % and 22 % are calculated, which cannot be seen from Fig.9.
(8) The conclusion needs to be optimized and simplified as much as possible.
(9) The English grammar and professional nouns in the article need to be improved. It is recommended to ask native English speakers to modify and improve them.
The description of Results and Discussion in the third section is not accurate enough, and the logic is somewhat confusing and needs to be carefully modified.
Author Response
see attachment

Reviewer 2 Report (Previous Reviewer 3)
line 294-295 and 297-298 and 17
you write that according to the standard presented in literature no. 35 is assigned to class B25 with designation M350
but there is no such marking in this standard of the European Union, concrete classes in this standard EN126 are written differently, look and write according to EN not according to GOST
Line 111: N - not a class, but a normal hydration reaction
Line 123: Review the units of measurement for density
there is no normal density - there is particle density or bulk density
and still I did not understand why quick-hardening concrete was mixed using cement with a normal hydration reaction and not with a rapid hydration reaction
line 281-283: there are electron microscope studies, but the description of the methods does not describe how these studies were performed, and the size measurement figures in Figure 3, 5 and 8 are not visible at all
Author Response
Dear Reviewer, thank you very much for considering our paper! We carefully revised the paper according to your comments and suggestions, as well as responded to each of them below. We really hope that the revised version of the paper meets your expectations.
Please see the attachment.

Reviewer 3 Report (Previous Reviewer 4)
The paper has improved since the last submission, but further improvements are needed. The language and the presentation needs improvement.
In the introduction you write: "The goal of this study was to find the utmost composition of a rapid hardening concrete with high performance properties (compressive strength and frost resistance)." What do you mean with utmost composition and are you considering all types of rapid hardening concretes available?
The introduction is very narrow and does not provide sufficient background to the subject or put the study in a broader context. One sentence in the introduction is not suitable for an introduction of a scientific paper "In our opinion, the advantages of obtaining high-quality modified RHCs are......"
The materials and methods section is very messy and needs to be rearranged for a better structure.
The results needs to be improved with a more thorough analysis and discussion of the meaning of the results. This is also needed as a base for the conclusions.
The conclusions are not based on any supporting discussion or analysis.
Author Response
Dear Reviewer, thank you very much for considering our paper! We carefully revised the paper according to your comments and suggestions, as well as responded to each of them below. We really hope that the revised version of the paper meets your expectations.
Please see the attachment.

Round 2
Reviewer 3 Report (Previous Reviewer 4)
Thank you for revising the paper according to my comments in the previous two rounds. The paper has improved in both rounds and is now ok according to my opinion.
Author Response
thank you
This manuscript is a resubmission of an earlier submission. The following is a list of the peer review reports and author responses from that submission.
Round 1
Reviewer 1 Report
There is no novelty in manuscript, but however, you should have potent more the importance of mixtures examined. Please find one of the published manuscripts about topic that you examined and write yours similar to it. Guidance for writing is given in word file as comments.

Author Response
Dear Reviewer,
Thank you for considering our paper. We really appreciate your comments and suggestions towards improving the quality of the paper.
We have carefully edited the article and would appreciate its reconsideration.
Please see the attachment.
Thank you in advance!
Best regards,
Authors

Reviewer 2 Report
This manuscript presents studies on the strength of rapidly hardening concrete using three main chemical modifiers and discusses frost resistance tests of concrete samples with optimized combinations under severe environmental conditions to evaluate their durability. The topic studied in this manuscript, especially the frost resistance test, is interesting, but the manuscript is not suitable for publication in the current form. Finally, the overall writing needs to be improved.
1. The introduction of this manuscript is too long with flight of ideas, lacking the latest literature review related to this study, which makes the design of the article confused. In other words, the authors do not summarize the studies on the effect of established chemical additives on rapid hardening concrete. The technical thinking of a good scientific paper should be very clear and rigorous.
2. The main chemical components of Portland cement CEM I 52.5H should be provided.
3. The gradation of fine aggregate and coarse aggregate will affect the final strength. Therefore, the gradation curve is recommended.
4. All the graphics in this manuscript are not at the level of a scientific or technical paper; your drawing skills need to be improved. For example, Figs. 2 and 3 are meaningless. The author should pay attention to the rheological properties, setting time, slump and other test indexes of the fresh mixture.
5. How do you achieve the freezing of samples in the air (-18 ± 2 ºC)? What kind of refrigeration equipment was used?
6. How many samples did you use for a particular strength test? The variability of concrete should be paid attention to. The author should show the standard deviation in Figs. 4-8.
7. This manuscript appears to be an experimental report, and the authors lack an in-depth analysis of the test results. Nor does it reveal the mechanism of the effect of modified chemicals on the strength of hardened concrete with the help of scanning electron microscopy and other means.
8. The conclusion should be condensed.
Author Response

(The authors gave the same response as above.)

Reviewer 3 Report
The abstract uses too many instances of the word “of” too close together. Consider rewording or splitting up the first sentence of the abstract.
Line 29: The word “of” is used too many times. Consider making it “a real possibility of accelerating hardening in RHC” instead.
Line 30: The word “of'' is used too many times. Consider making it “positive effect in kinetics acceleration of strength gain” instead.
Keywords: A full stop is missing at the end of the keywords section. Also, “modifiers” is a rather vague keyword, consider specifying the type of modifiers.
Lines 43 and 109: It should be “work” instead.
Line 59: It should be “contributing” instead.
Lines 72-74: Sentence seems badly worded. Consider using “Also a great deal of researchers’ attention is directed to different types of polycarboxylate hyperplasticizers with different structures of the chemical molecule.” instead.
Line 83: The word “of” is again used too many times. Consider making it “from the authors” instead.
Line 144: The colon (:) symbol seems to be used but only one material is listed in the next line. Consider replacing it with a dash (-) instead.
Lines 198-199: It should be “there was also data used from Table 5”.
Figure 5: Some of the values of orange columns seem to be missing, presumably gone out of frame.
Lines 333-334: It should be “concretes allow to significantly accelerate”.
Lines 370-372: It should be “This confirms that the main task of the current study aimed to increase durability and service life of RHC using modifiers is solved successfully”.
386 line and all and Table 1: GOST write what regional interstate standard, CIS countries?
Line 110: correct the chemical formula
Line 145: What does it mean „of "Standart Cement" plant“? write in which factory the cement was made and explain what the letter H means CEM I 52.5H
Lines 149-158: what is the granulometric composition of sand?
Lines 164-167: what is the fineness of microsilica?
Figure 2: how was the even distribution of the additive ensured?
Table 4: correct the chemical formulas
Lines 105-111: the literature review should mention that calcium chloride admixture can only be used in non-reinforced concretes, as the use of chlorides is prohibited
Table 5: correct the chemical formulas
Line 427: it is a european union standard
Line 195: [32] not listed
Line 200: correct the chemical formula
Table 5: at what humidity were the samples cured?
Line 230: correct the chemical formula
Lines 243-245: the formula contains a correction factor due to cellular concrete, nowhere in the article is it noted that cellular concrete was produced, if ordinary concrete was produced, this factor is not needed.
Separately describe the test methods in one chapter.
Line 253: write according to which document the concrete was classified
Tables 6-8 repeat the results with Figures 4-7, leaving only one place
Line 337: B25 according to which document was this classification made? what does it mean?
Author Response

(The authors gave the same response as above.)

Reviewer 4 Report
The paper presents results from tests of accelerators and modifiers for faster strength development and frost resistance. The additives used for rapid hardening concrete (RHC) was microsilica, polycarbolate esters and calcium chloride. The tests showed good results for a combination of microsilica and calcium chloride, with a favourable level of accelerated strength development and frost resistance for a concrete mix containing 30% calcium chloride.
The paper appears a bit confusing and there are some aspects of the paper that needs to be thoroughly improved, mainly concerning the description of the test methods, which in the current state are not described sufficiently. I can for example not find sufficient information about how many and what type of specimen that were tested, in Table 10 you mention 150 mm and 100 mm cubes. I can not find details on how the frost resistance tests were carried out. In some part you mention a temperature of 20±5°C and in some other part there is a specification of 20±2°C. The effect of the temperature is probably a very big factor of uncertainty, since the strength development in 15°C is quite slower than in 25°C.
The aim of the paper is also a bit confusing, since there are several different descriptions of the aim in different parts of the paper. For example, one purpose is defined as (line 95): “The purpose of the present study was to select optimal methods of accelerating hardening without deteriorating the performance characteristics and reducing the cost of popular classes of rapid hardening heavy concrete by using complex modifiers and 2nd generation hyperplasticizers with targeted properties”. The paper does however not select any optimal methods, it studies the effect of three additives, and no cost perspective is investigated.
There are also some confusing statements throughout the text. For example (line 37): “In normal thermal and moisture conditions, concrete reaches its ultimate strength within 28 days of curing [1]”.
The methods and materials section seems to include some results that should be moved to the results section of the paper and Figure 1-3 are perhaps a bit irrelevant. Table 5-12 seems to contain the same information as Fig. 4-9 and the tables are a bit messy.
In order to be accepted, the paper needs a thorough revision with rearrangement of the structure, a better method description, better use of tables and figures and an aim that is consistent and actually addressed in the paper.
Author Response

(The authors gave the same response as above.)
